# Survivorship and yield of a harvested population of *Forsteronia glabrescens*

**Demetrio Luis Guadagnin** [ORCID]*, **Paulo Vinícius Fernandes Barradas**

Department of Ecology, Laboratory of Wildlife Ecology and Management, Universidade Federal do Rio Grande do Sul, Porto Alegre, RS, Brazil

* dlguadagnin@gmail.com

## Abstract

The exploitation of non-timber forest products may be an opportunity to reconcile the utilization of biological resources with biodiversity conservation. In Southern Brazil, the exploitation of liana stems for handicraft makes up an important part of the income of indigenous Kaingang people. In this study we evaluated the effects of stem harvesting on the survivorship of *Forsteronia glabrescens* Müll.Arg, the most exploited liana species in the region. We marked and monitored the survivorship, sprouting, changes in stem diameter and resource yield in control and harvested plots with two different resting times–six and twelve months. We associated variables of interest with individual attributes, harvesting regime and vegetation descriptors through linear mixed modelling. Survivorship and resource yield were lower in the harvested groups than in the control group, although the mean stem diameter was higher. Plants with larger stem diameter presented higher survival odds. Either six or twelve months of resting between harvests were not sufficient to recompose the yield and compensate mortality. Harvesting twice a year increases yield but reduces survivorship. Our results point that the sustainable exploitation of *F. glabrescens* require either large areas, low pressure or resting periods longer than the ones we tested.

## Introduction

The sustainable use of biological resources is regarded as a way of conserving while valuing biodiversity [1], but many populations of exploited plant and animal species are in decline by the combined effects of various threats, including overexploitation [2–5]. Information on demographics are important subsidies for guiding management strategies [2, 6]. By definition, a positive growth rate ($\lambda > 1$), that is, natality greater or equal to mortality, is a basic requirement for sustainable exploitation, so that the number of exploited individuals can be replenished [7]. Survivorship is one of the most informative population parameters regarding the fate of individuals in exploration scenarios, and changes in this parameter may affect population dynamics and structure [2]. Non exploited, stable populations have mortality equal to natality and so the abundance is reduced at the beginning of exploitation, giving rise to a positive growth rate which is the sustainable yield. Mortality imposed by exploitation influences the population dynamics in a continuum of situations delimited by two extremes:

**Data Availability Statement:** All relevant data are within the paper and its Supporting Information files.

**Funding:** Paulo Barradas had a scholarship from the Coordenação de Aperfeiçoamento de Pessoal

de Nível Superior (CAPES) for the Programa de Excelência Acadêmica (PROEX). We declare that the funders that supported a scholarship to Paulo Barradas had no role in study design, data collection and analysis, decision to publish, or preparation of the manuscript.

**Competing interests:** The authors have declared that no competing interests exist.

compensatory and additive mortality [8]. In the first case, the removal of some individuals alleviates competition, increasing recruitment. In the second case, the mortality due to harvesting individuals is added to that caused by other factors, therefore decreasing recruitment.

The compensation between density, growth and survivorship (self-thinning) is regarded as a common pattern in plants [9–12]. The exploitation of many NTFPs involves removal of just a part of each individual, hence its death is not certain, but may occur as a delayed consequence of multiple or successive stressful harvest events [5, 13]. There may also be a trade-off between vegetative recruitment and reproductive capacity [10, 14].

Lianas are characteristic components of forests [15, 16] and raw materials for handicrafts [17]. Their attributes make them good raw material for basketwork and other handicraft artefacts. The high growth rates of some species, and the rapid colonization of forest edges and early successional phases cause them to be abundant elements in secondary forests and altered fragments [16–18]. Many lianas have great capacity of regenerating after damage, so that a new branch quickly replaces a damaged branch [19]. In many species of lianas several stems may be ramets of the same genet, resulting from vegetative reproduction through stolons or rooting of stems [19–21]. Despite attributes favouring their increase in abundance in primary and secondary forests [16–18], some species are undergoing population declines due to over-exploitation [5, 22–24] or other processes [25].

In South Brazil, the production of liana handicrafts by the Kaingang natives, sold in local, urban markers, is nowadays a central activity of family-based productive modes, webs of sociability and sources of income [26]. There are at least 37,000 Kaingangs in South Brazil, about their halves in cities, where they live mainly from the sale of crafts and NTFPs [27, 28]. The difficulty of accessing distant areas and restrictions on access to public and private lands impose a high level of exploitation on some localities. *Forsteronia glabrescens* Müll.Arg. (Apocynaceae) is the most abundant liana species in the region and the most used species by Kaingangs [26]. *F. glabrescens* is a multi-branched, twining, heliophyte liana that develops well in the secondary shrublands and forest edges and interiors in subtropical South America. It can grow vegetatively, producing stems that root once spreading through the ground, giving rise to new ramets. The exploitative practice of the Kaingang basically includes cutting the stems at the base of a genet and than pulling both the ground and climbing parts of the stem. They adopt a strategy to avoid overexploitation–break periods of six to twelve months, during which some areas are left for regrowth. In this paper we investigate how the exploitation of *F. glabrescens* affects its survivorship and yield. We reproduced experimentally, under field conditions, the stem harvesting practised by the Kaingang natives to test if survivorship and yield differ between exploited and unexploited plots. We also evaluated how different intervals between harvests (6 or 12 months) affect the survivorship and regrowth of stems.

## Methods

The field experiment was carried out in the Morro São Pedro Wildlife Refuge (30°10'27" S, 51° 6'12" W), southern Brazil. The field research was authorized by the Secretaria Municipal do Meio Ambiente da Prefeitura Municipal de Porto Alegre under the Permit no. 308/07, in accordance with the *Instrução Normativa ICMBio n° 03/2014*. The climate is subtype Cfa, with average annual temperature of 19,5°C, and the average annual precipitation is 1330 mm. The studied forest is in advanced stage of succession and shows no signs of liana exploitation or any other human interference at least in the last 50 years [29].

We sorted a random start and direction within the forest and systematically allocated 15 plots of 5 m x 5 m (total of 350 m$^2$), 15 meters apart, at a minimum distance of 10 m from the forest edge. The plots were assigned randomly into three levels–five control plots and nine

treatment plots: four plots subjected to one annual harvest and five subjected to two semi-annual harvests. We labelled all individuals of *F. glabrescens* with numbered tags and measured the diameter of the stems at 30 cm from the ground with a pachymeter [30]. The treatments consisted of harvesting the stems simulating the exploitation practice of the Kaingangs [26]– we cut the stems about 30 cm from the ground and pulled down their aerial portion. We excluded from counting and harvesting young sprouts with length <1.0 m or diameter <0.2 mm, because they are not suitable for making handicrafts [26]. We measured the stem diameter and the harvested length, which is less than the total length, since stems which are very intertwined with vegetation break during harvest. Every six months we recorded the number of surviving stems and new sprouts and took new diameter measures. We installed the experiment in December 2014 simulating the harvest in the nine treatment plots. We reapplied the harvest after six months (June 2015) in five plots and in all nine ones after one year (December 2015).

We considered five response variables: 1) Stem density: the total number of stems found within each plot; 2) survivorship: the binary individual fate of stems after the first six months, the last six months, and over one year; 3) Change in mean stem diameter: the mean variation in the diameter of the stems over the three instances; two variables to estimate the yield: 4) Change in the mean harvested length: the mean variation in length of stems obtained from every individual at each harvest, accumulating the two harvests of the semi-annual group at the end of the study; 5) Yield per plot: the accumulated length of stems harvested in every plot after one year divided by the number of individuals alive at each harvest, averaged among the plots under the same treatment. We considered all new sprouts emerging from soil and not clearly linked to a marked stem as recruits, recognizing that they could be either new ramets or genets, and calculated an annual recruitment rate as the number of new sprouts divided by the number of stems marked at the start of the study and alive in the plot after one year.

We considered three groups of explanatory variables plausibly related with yield, growth and survivorship. 1) The individual measure of *F. glabrescens* diameter to represent the effect of the initial plant condition. 2) Two nested, categorical variables to represent the plant allocation in the experiment: a) "new" = plant born during the experiment, "control" = unharvested plant and "harvested" = plants that undergone stem harvest; b) the harvested plants, classified into two groups: annual–one harvest every 12 months; semi-annual–one harvest every six months. 3) We used four variables describing the vegetation within each plot as random effects in order to control for the effect of potential differences in the forest stands on the abundance and fate of lianas and the availability of support trees and canopy cover: tree density, mean and maximum tree diameter at breast height (DBH), and tree DBH standard deviation. These DBH measures can account for potential effects of the variability in the availability of support trees and in the canopy cover, known to affect the abundance and growth of lianas [31–34]. All numerical explanatory variables were standardized prior to analysis.

Our routines of analyses follow [35]. We established a global model for each parameter of interest (survivorship, growth, yield and density), including all explanatory variables. We used GLMMs with binomial distribution and logit link function for survivorship; LMMs for change in stem diameter; LMs for yield per plot and change in mean stem length; and GLMs with Poisson error distribution and logarithmic link function for density models. We used the identity of each plot as a random variable in the construction of the mixed models (GLMMs and LMMs). The parameter estimation was performed through multi-model inference based on the set of all best models with $\Delta AICc \leq 2$ [36]. All analyses were performed using the packages bbmle [37], mgcv [38] and lme4 [39] on R [40]. We checked the performance and fit of the global model examining effect sizes, distribution of residuals [41] and the marginal and conditional $R^2$ [42].

## Results

We identified and tagged 167 individuals of *F. glabrescens* (S1 Table). The mean density was 9.4 stems per plot of 25 m$^2$ (min = 3, max = 19), corresponding to 4,743 stems/ha (3,761 stems < 1cm; 982 stems > 1 cm). The mean initial diameter was 0.70 cm (min = 0.05 cm, max = 3.39 cm; Fig 1). The mean diameter on the treatment plots decreased to 0.22 cm after one year. The density of *F. glabrescens* was positively related to tree density (Estimate = 0.22 ± 0.24 SE), mean DBH of trees (0.18 ± 0.11) and maximum DBH of trees (0.18 ± 0.10) (Table 1). The control group showed minor variations in stem diameter along the year (maximum of 5%).

The control group had a higher survivorship rate (93.75%) than the harvested groups (66.66%) in the first six months (Fig 2). Survivorship was lower in the first than in the second period. After one year, the survivorship rates were 80.70% in the control group, 55.81% in the annual harvest and 53.19% in the semi-annual harvest. The cutting of stems reduced the survivorship in both treatments in the first period and, by consequence, in the whole period (Table 1). Stems with larger diameter presented higher survivorship in the last six months and in the annual harvest. Tree density and mean tree DBH were positively related to survivorship in all periods, but the maximum tree diameter had a negative relationship with survivorship.

Along one year the control group didn't increase in diameter, while the extracted stems increased 0.19 cm in average. While the group harvested only once showed a vigorous regrowth only in the first six months, the group harvested twice regrew up vigorously after each harvest (Fig 3). The mean stem diameter increased with the cutting of stems in the first period, and decreased with the initial stem diameter (Table 1), so that the investment in radial growth was greater in the individuals with lower initial diameter. Yield increased with the

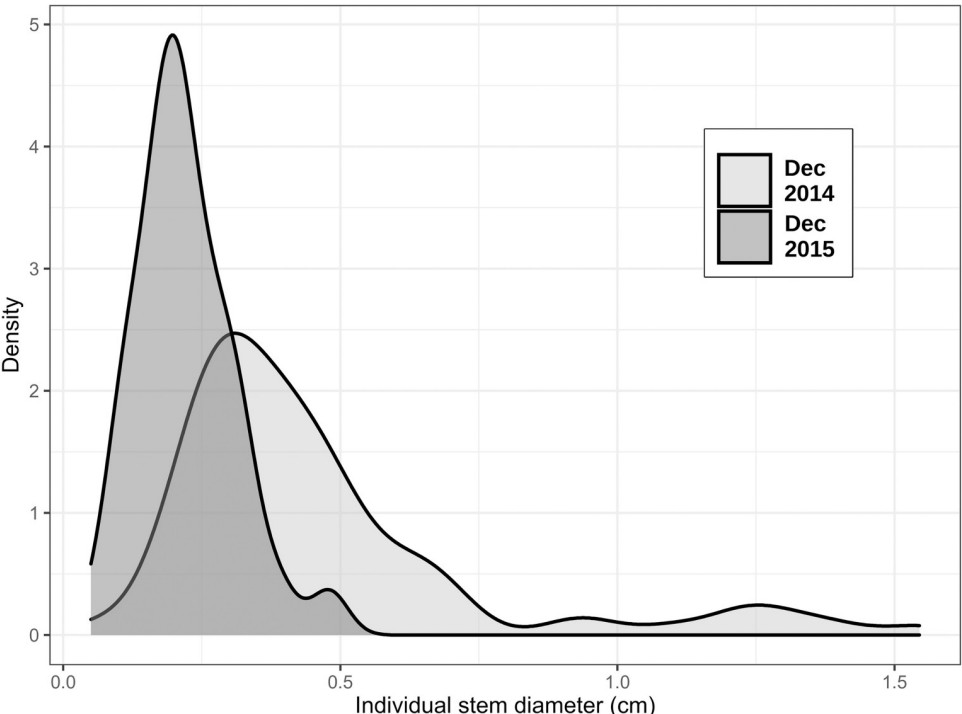

**Fig 1. Change in the population structure of *Forsteronia glabrescens* population one year after being submitted to stem harvesting.**

**Table 1. Fit and summary of estimates (multi-model averaging) of effects (± standard errors) of *Forsteronia glabrescens* attributes and forest stand factors on population parameters and harvest yield.**

| Parameter estimates | Density of *F. glabrescens* | Survivorship | | | Change in mean stem diameter | | | Harvested length (log) | Yield per plot |
|---|---|---|---|---|---|---|---|---|---|
| Period of time | Annual | First period | Second period | Annual harvest | First period | Second period | Annual harvest | Annual | Annual |
| Global model Conditional $R^2$ | 0.456 | 0.265 | 0.728 | 0.473 | 0.321 | 0.431 | 0.736 | 0.160 | 0.567 |
| Stem diameter | | | 2.55±1.47 | 1.31±0.41 | -0.07 ±0.03 | -0.13±0.05 | | | |
| Stem cutting | | -2.05 ±0.69 | | -0.81±0.61 | 0.26±0.07 | | 0.72±0.13 | -0.48±0.35 | -1084.5 ±531.0 |
| Annual harvest | | | | | | | | | -1763.2 ±540.8 |
| Semi-annual harvest | | | | | | | | | -1112.3 ±474.8 |
| Density of *F. glabrescens* | | | | | | | | | 331.8±206.7 |
| Density of trees | 0.22±0.10 | | | 0.49±0.39 | | | | | 486.4±230.9 |
| Mean DBH of trees | 0.18±0.11 | 0.18±0.22 | | 0.80±0.41 | | | 0.19±0.11 | | |
| Maximum DBH of trees | 0.18±0.10 | | | -0.66±0.43 | | | | | |

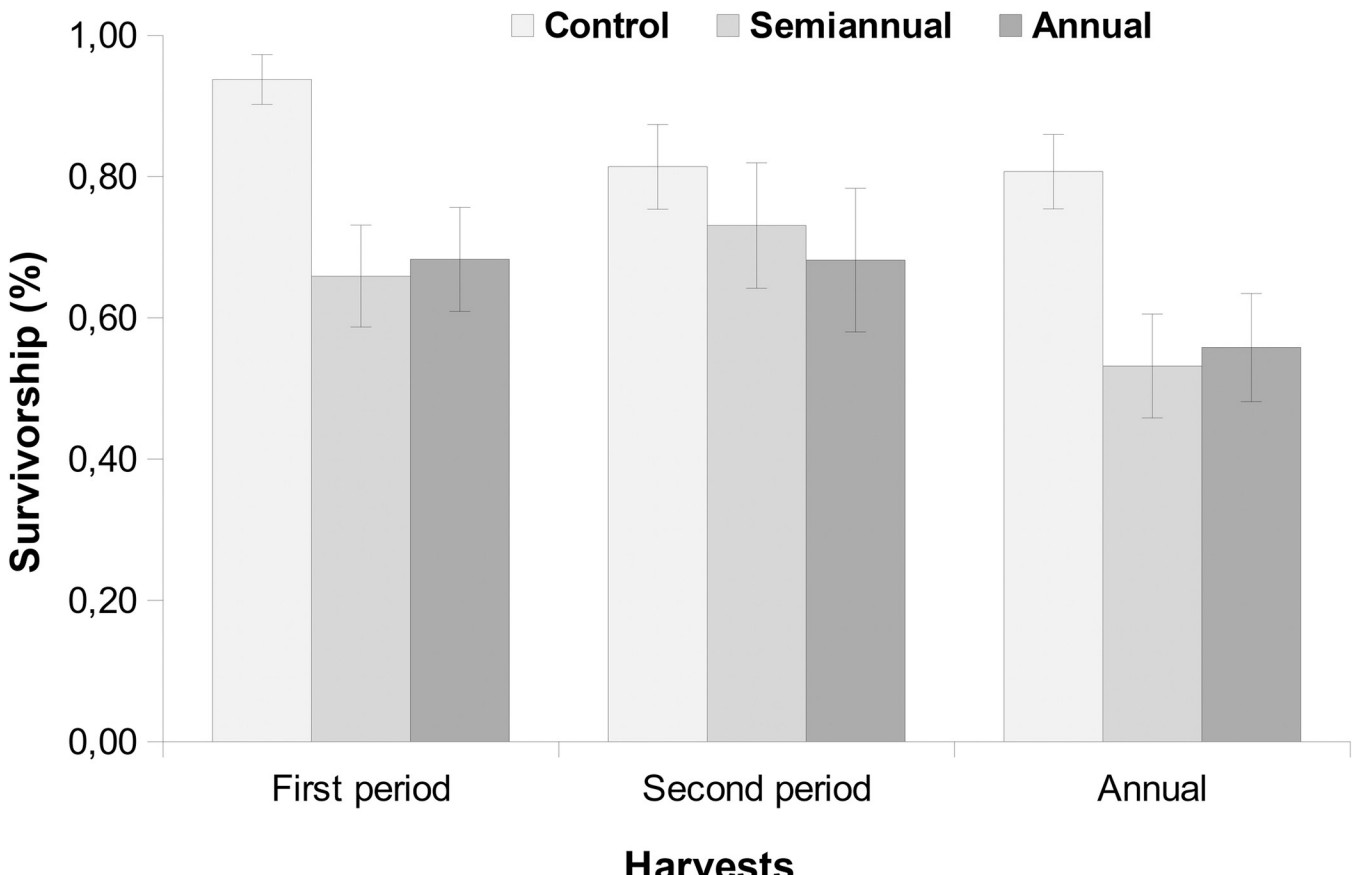

**Fig 2. Survivorship of *Forsteronia glabrescens* (mean ± standard error) in two harvesting periods of six-months each and after one year, comparing stems unharvested and submitted to harvesting once and twice a year.**

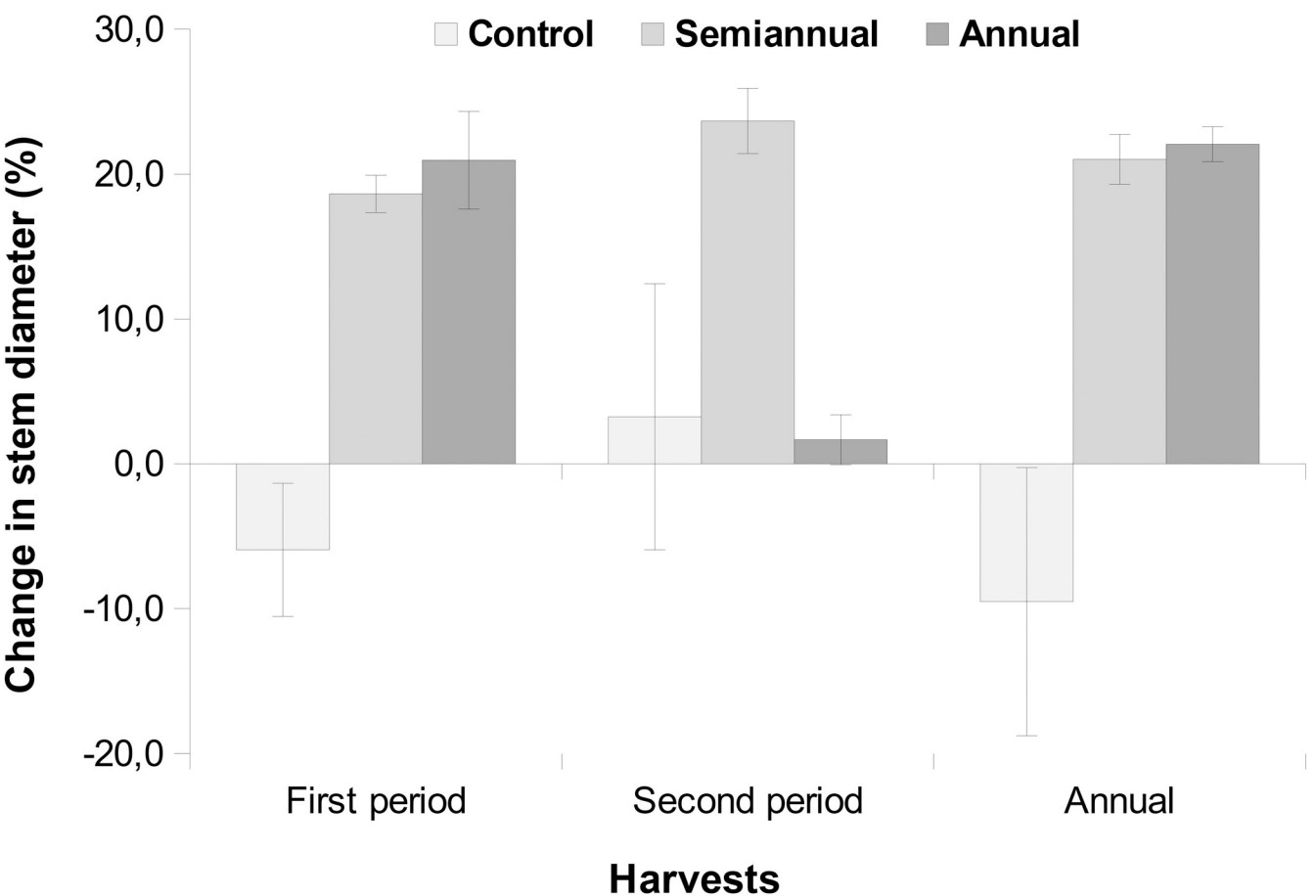

**Fig 3. Change in mean (± standard error) stem diameter of *Forsteronia glabrescens* surviving after six months and one year of harvesting and unharvested, control group.**

density of trees and *F. glabrescens* and decreased with the cutting of stems. Considering the two levels of treatment as one, the average increase in diameter was 0.30 cm in one year.

The annual recruitment rate, calculated at the end of the study, was 17.7% in the control group, 17.6% in the annual harvest group and 11,3% in the semi-annual harvest group.

At the end of the study, the mean ± standard deviation of the harvested stem's length of the control group, was 278.3 ± 203.4 cm, while the treatment groups averaged together 199.9 ± 130.8 cm. The growth in length of the harvested groups after one year was in average 40% lower than their initial length (Fig 4). The total yield of the group that was explored at six months was 11% greater than the group harvested only once per year (see also Table 1).

## Discussion

In this work we demonstrate that, compared with unexploited plots, the survivorship of *Forsteronia glabrescens* decreases in harvested plots, that individuals with thicker stems present a greater chance of survivorship and that harvesting twice a year increases yield but reduces survivorship. We also have shown that the yield in managed populations is lower than that obtained in populations not previously exploited and that both six and twelve months break periods are insufficient for recovery.

The exploitation of *F. glabrescens* seems to follow the self-thinning principle of compensatory growth following a disturbance, under-compensating the stem cutting with secondary

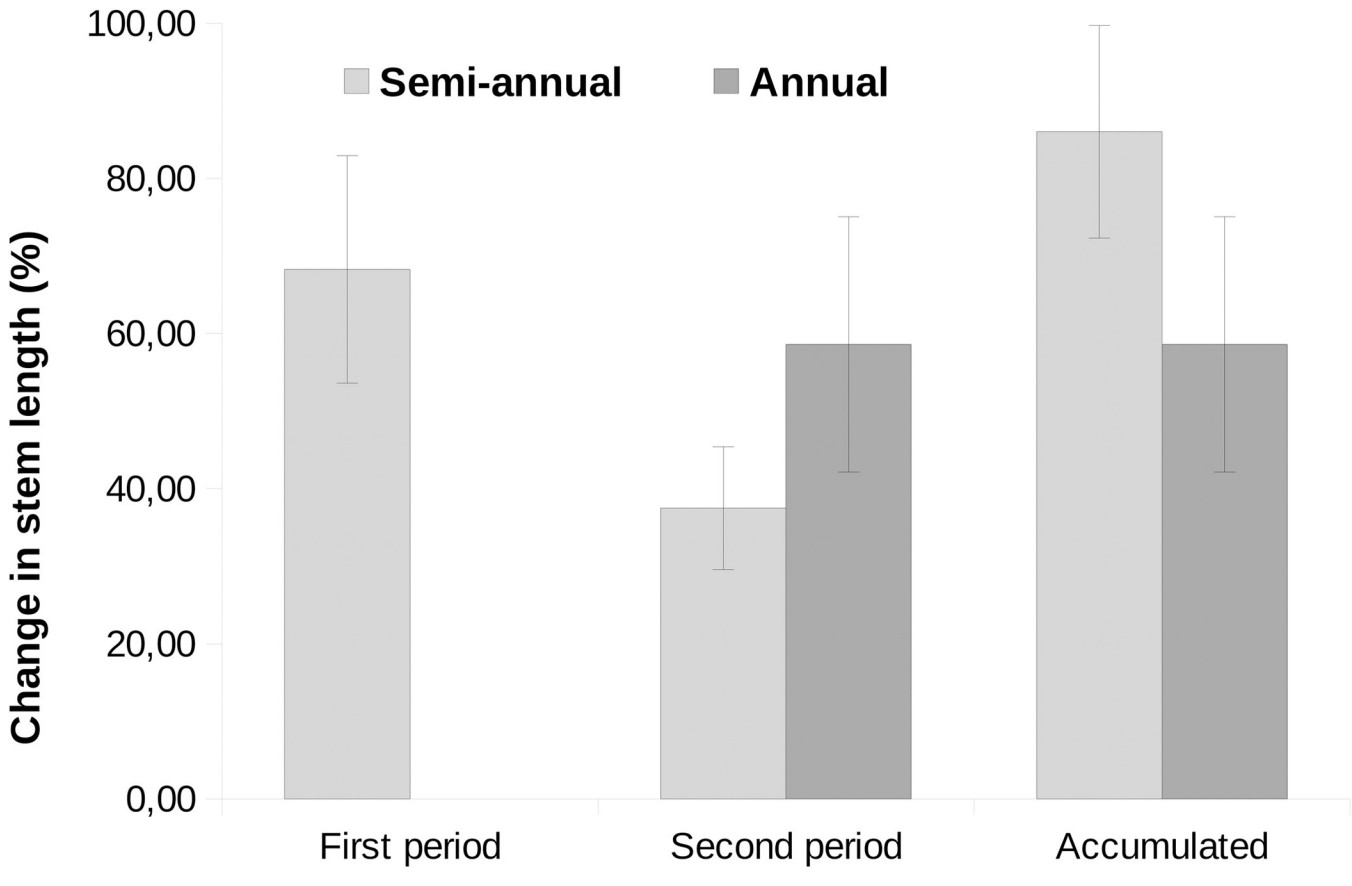

**Fig 4. Change in mean stem length of *Forsteronia glabrescens* (mean ± standard error) in six-months periods and accumulated after one year, comparing stems submitted to harvesting once and twice a year.**

growth and new sprouts up to a stand biomass bellow that of the control stands [11, 12]. We did not find significant differences in survivorship and regeneration between the first and second crops. However, in the first harvest, cutting the stems reduced the survivorship from 93.87% in the control plots to 66.66% in the extractive plots. Other cases of exploitation of perennial, understory NTFPs also noted that the cutting of aboveground stems reduced the survivorship [43]. Several other figures reinforce the under-compensation of the population at the experimented conditions. The individual survivorship probability increased by 29.8% for each centimetre of diameter. In terms of total yield, the recruitment and growth of stems along the period studied does not compensated the death of previously harvested plants. The average length of harvested stems was approximately 37% lower in the plants already submitted to exploitation than those obtained from plants not previously harvested, and 40% lower in a second harvest. The fact that individuals were cut once or twice in a year had a small effect over this variable. Considering the total amount harvested per study plot, cutting the stems also had a negative influence, that is, previously exploited plots yielded a total amount of resource approximately 57% lower than areas without previous exploitation. Under-compensation is an expected response to herbivory-damaged perennial plants, which tend to reallocate energy in

inedible, bellow-ground storage organs first, for future growth after de acute disturbance [10], which is a situation similar to an exploitation bound.

Decreased size and altered population structure are expected consequences in natural populations subjected to exploitation [6, 44, 45]. Mortality caused by exploitation can be either compensatory or additive when it removes individuals [8], but the effect on survivorship varies when only a portion of the plant is harvested, and can go from compensatory to additive as harvesting intensity increases [46]. The population changes can be masked until the depletion of energy reserves [47, 48], which means that the initial yield of an unexploited population does not necessarily reflect long-term, sustainable yield. It is expected that, as the population structure is altered by exploitation, the progressive elimination of larger individuals decreases the survivorship and the population growth rates, reducing the yield achievable by further exploitation [2]. The regenerative capacity after the first harvest may be greater than that which the population will be able to maintain after consecutive harvests [2, 12, 43, 49].

The density of available trees influenced positively the individuals' survivorship and the total yield, as well as the density of *F. glabrescens*, which is probably related with a greater availability of potential host trees [32, 50]. The abundance of lianas in tropical forests is usually greater in clearings and open canopy areas that allow greater light entry, which is associated with the aggregated distribution of many species [31, 32]. This is not the case for *F. glabrecens*, and other species which present random distribution [26, 32, 34]. The availability of climbing trees may be the main factor modulating the abundance of lianas bellow a certain light threshold [31, 51] or depending on stand age and the size of the host trunks [33]. Alternatively, this is a plausible outcome if the nature of the relationship between trees and lianas is parasitic rather than competitive [52]. If the dominant population outcome is beneficial to lianas and harmful to trees, the relationship could be better described as parasitic [52]. This outcome is probable when the relationship is facultative and services, instead of food, are demanded by the parasite. This is a largely unexplored subject.

We analysed the effect of harvesting in two sequential harvests along one year, comparing two different regimes and controls. We acknowledge that long-term longitudinal studies may reveal other effects on population structure and trends and succession. Although short to confirm general demographic parameters assuming that mean values of survivorship, growth and crop yield vary among years, we are confident that these variation cannot reverse the effects of harvesting we detected on the estimated parameters. We are also confident that our study is sufficient to affirm that harvesting increases mortality and reduces density and crop yield, and that harvesting twice a year has greater effects on population parameters than harvesting once a year. This decrease in density is an expected condition of the exploitation of any biological resource, but the increase in mortality of previously harvested stems means a progressive worsening of condition of the genet. One plot of annual harvest had some labels lost or removed, so leading to a somewhat unbalanced design. Although, GLMM is robust against unbalanced groups provided that there is no heterogeneity of variances [53, 54].

We treated as individuals the ascending stems associated to trees, not knowing to what extent there may be subterranean connections between the tagged stems. Future studies should broaden the knowledge of the species basic biology, in particular the understanding of its sexual and vegetative reproduction, phenology, the distinction between genets and ramets, germination and sprouting. However, we consider our results to be consistent and representative for the populations of *F. glabrescens* in the Atlantic, secondary forests of southern Brazil where most of the exploitation occurs. Our results rise concerns about the sustainability of stem harvesting of the studied species and we therefore call for continued monitoring of harvested populations.

Our work demonstrates that the exploitation of *F. glabrescens* in southern Brazil, either at six months or one year intervals, leads to increased mortality rate, reduced mean size of ramnets, yields smaller than expected in previously unharvested areas, and a risk of even smaller yields if the exploitation goes on continuously. Either larger areas, lower pressures, or longer resting periods may be necessary to exploit *F. glabrescens* sustainably.

## Supporting information

**S1 Table. Size and fate of *Forsteronia glabrescens* Müll.Arg ramets experimentally managed in the São Pedro Municipal Reserve, Porto Alegre, South Brazil, 2014–2015.** (TXT)

## Acknowledgments

The authors thanks to the São Pedro Wildlife Refuge for upholding the experiment and providing logistic support and facilities

## Author Contributions

**Conceptualization:** Demetrio Luis Guadagnin.

**Formal analysis:** Paulo Vinícius Fernandes Barradas.

**Funding acquisition:** Demetrio Luis Guadagnin.

**Investigation:** Demetrio Luis Guadagnin, Paulo Vinícius Fernandes Barradas.

**Methodology:** Demetrio Luis Guadagnin.

**Project administration:** Demetrio Luis Guadagnin.

**Writing – original draft:** Paulo Vinícius Fernandes Barradas.

**Writing – review & editing:** Demetrio Luis Guadagnin.

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
