## [Decision Letter · Decision Letter 0]

5 Oct 2021

PONE-D-21-21671Survivorship and yield of a harvested population of Forsteronia glabrescens .PLOS ONE

Dear Dr. Gaudagnin,

Thank you for submitting your manuscript to PLOS ONE. After careful consideration, we feel that it has merit but does not fully meet PLOS ONE’s publication criteria as it currently stands. Therefore, we invite you to submit a revised version of the manuscript that addresses the points raised during the review process.

We look forward to receiving your revised manuscript.

Kind regards,

Ricardo Alia, PhD

Academic Editor

PLOS ONE

Journal Requirements:

Additional Editor Comments (if provided):

The paper present some interesting results on population of Forsteronia glabrescens and it is of itnerest for the sustainable managment of this species. There are some comments by the two referies that should be taken into consideration for the authors, as they can improve the quality of the paper. You should specify if the data is included in any public repository as you mention that the data is accessible without restrictions. Please, check the final part of the conclusion (Either larger areas, lower pressures, or longer resting periods are necessary to exploit F. glabrescens sustainably) as it seems that this conclusion is not directly extracted from your data. I also think that your study is rectricted to one location, and therefore you should mention any caveats to extrapolate your results to the management of the species in general. Please check in line 144 after packages if the name of one package is missing before the first reference.

Reviewers' comments:

Reviewer's Responses to Questions

**Comments to the Author**

1. Is the manuscript technically sound, and do the data support the conclusions?

Reviewer #1: Yes

Reviewer #2: Yes

2. Has the statistical analysis been performed appropriately and rigorously? 

Reviewer #1: Yes

Reviewer #2: Yes

3. Have the authors made all data underlying the findings in their manuscript fully available?

Reviewer #1: Yes

Reviewer #2: No

4. Is the manuscript presented in an intelligible fashion and written in standard English?

Reviewer #1: Yes

Reviewer #2: Yes

5. Review Comments to the Author

Reviewer #1: The Manuscript presents important elements to guide sustainable extractivism.

References are out of date

o Uneven number of parcels could not have affected the results? what was the authors' strategy to resolve this situation?

Reviewer #2: The manuscript “Survivorship and yield of a harvested population of Forsteronia glabrescens” submitted to Plos One describes the effect of harvesting on survivorship of a liana species in forest area of Brazil. It is well written, the ideas are clear, and the interest part is that encompass several subjects (ethnobotany, ecology, and forest management). The following observations may improve this manuscript:

1. First paragraph in the Introduction is too long. I suggest the authors to divide the ideas in two shorter paragraphs.

2. The second paragraph introduces the idea that the population of some liana species are declining due to overexploitation. It would be necessary to add a sentence about increasing liana abundance in the last decades:

Schnitzer, S. A., & Bongers, F. (2011). Increasing liana abundance and biomass in tropical forests: emerging patterns and putative mechanisms. Ecology letters, 14(4), 397-406.

Schnitzer, S. A., Mangan, S. A., Dalling, J. W., Baldeck, C. A., Hubbell, S. P., Ledo, A., ... & Yorke, S. R. (2012). Liana abundance, diversity, and distribution on Barro Colorado Island, Panama. PloS one, 7(12), e52114.

Ingwell, L. L., Joseph Wright, S., Becklund, K. K., Hubbell, S. P., & Schnitzer, S. A. (2010). The impact of lianas on 10 years of tree growth and mortality on Barro Colorado Island, Panama. Journal of Ecology, 98(4), 879-887.

3. I recommend adding general information about liana demography in the Introduction. For example, relationships and trade-offs between growth and mortality, and shade-tolerance. Some papers on this regard:

Gianoli, E., Saldaña, A., Jiménez‐Castillo, M., & Valladares, F. (2010). Distribution and abundance of vines along the light gradient in a southern temperate rain forest. Journal of Vegetation Science, 21(1), 66-73.

Ceballos, S. J., & Malizia, A. (2017). Liana density declined and basal area increased over 12 y in a subtropical montane forest in Argentina. Journal of Tropical Ecology, 33(4), 241-248.

4. It would be necessary to explain why these variables were selected: tree density, mean and maximum tree diameter at breast height (DBH), and tree DBH standard deviation, and how these variables are related to this particular liana species. There are some information on the literature? For example, tree DBH standard deviation represents the availability of trees of different sizes to climb to the canopy?

5. What was the minimum diameter for a liana to be considered? This information is needed in Methods.

6. In Results change DAP by DBH.

7. In line 165, remove “The”.

8. Improve the quality of Figure 4.

9. Vines and ethnobotany are keywords but never mentioned in the text.

10. It would be useful to add a Discussion on methodological issues involve in this study. Particularly, about the differences between doing this study in an area not exploited before (a relatively protected area in late succession) and not in areas that were harvested several times in the past. This probably affected the Results.

6. PLOS authors have the option to publish the peer review history of their article (what does this mean?). If published, this will include your full peer review and any attached files.

Reviewer #1: No

Reviewer #2: No

---

## [Author Response · Author response to Decision Letter 0]

15 Dec 2021

Below, we address each specific comment made by the referees, addressed in the revised manuscript “Survivorship and yield of a harvested population of Forsteronia glabrescens” (PONE-D-21-21671 EMID bfb9dffccce607f0).

Responses to comments:

Journal Requirements:

→ We corrected the style of authorship, titles and fig captions, as well as the figure’s formats, as requested.

→ We included the statement that “ The field research was authorized by the Secretaria Municipal do Meio Ambiente da Prefeitura Municipal de Porto Alegre under the Permit no. 308/07 “ in Methods, lines 96-97.

→ We included a table with crude field data as Supporting Information. See answers #7 and #13

→ We included the statement that “The field research was authorized by the Secretaria Municipal do Meio Ambiente da Prefeitura Municipal de Porto Alegre under the Permit no. 308/07 “ in Methods, lines 96-97. See answer to question #2, above. 

→ We reviewed the references and in-text citation thoroughly according to the most recent guidelines

→ We added ten new references, attending questions #15 and #19, bellow:

1. Kor L, Homewood K, Dawson TP, Diazgranados M. Sustainability of wild plant use in the Andean Community of South America. Ambio. 2021;50:1681–1697. https://doi.org/10.1007/s13280-021-01529-7

2. Cunningham, AB, Brinckmann JA, Harter DEV. From forest to pharmacy: Should we be depressed about a sustainable Griffonia simplicifolia (Fabaceae) seed supply chain?,Journal of Ethnopharmacology, 2021;278:114202, https://doi.org/10.1016/j.jep.2021.114202

3. Zhou S, Lou Y-R, Tzin V, Jander G. Alteration of Plant Primary Metabolism in Response to Insect Herbivory. Plant Physiology. 2015;169:1488–1498. https://doi.org/10.1104/pp.15.01405

4. Li C, Barclay H, Roitberg B, Lalonde R. Ecology and Prediction of Compensatory Growth: From Theory to Application in Forestry. Frontiers in Plant Science. 2021;12:1352. https://doi.org/10.3389/fpls.2021.655417

5. Zhang T, Yu L, Man Y, Yan Q. Effects of harvest intensity on the marketable organ yield, growth and reproduction of non-timber forest products (NTFPs): implication for conservation and sustainable utilization of NTFPs.Forest Ecosystems. 2021;8:56. https://doi.org/10.1186/s40663-021-00332-w

6. da Cunha Vargas B, Grombone-Guaratini MT, Morellato LPC. Lianas research in the Neotropics: overview, interaction with trees, and future perspectives. Trees;2021;35: 333–345. https://doi.org/10.1007/s00468-020-02056-w

7. Mori H, Ueno S, Kamijo T, Tsumura Y, Masaki T. Interspecific variation in clonality in temperate lianas revealed by genetic analysis: Do clonal proliferation processes differ among lianas? Plant Species Biology. 2021;36:578–588. https://doi.org/10.1111/1442-1984.12348

8. Estrada-Villegas S, Hall JS, van Breugel M, Schnitzer SA. Lianas reduce biomass accumulation in early successional tropical forests. Ecology. 2020;101:e02989. https://doi.org/10.1002/ecy.2989

9. Lee ED, Kempes CP, West GB. Growth, death, and resource competition in sessile organisms. Proceedings of the National Academy of Sciences. 2021;118:e2020424118. https://doi.org/10.1073/pnas.2020424118

10. Ceballos SJ, Malizia A. Liana density declined and basal area increased over 12 y in a subtropical montane forest in Argentina. Journal of Tropical Ecology. 2017;33:241–248. https://doi.org/10.1017/S0266467417000153

Additional Editor Comments (if provided):

6. The paper present some interesting results on population of Forsteronia glabrescens and it is of itnerest for the sustainable managment of this species. There are some comments by the two referies that should be taken into consideration for the authors, as they can improve the quality of the paper.

→ Thank you for the recognition of our effort.

7. You should specify if the data is included in any public repository as you mention that the data is accessible without restrictions.

→ We included a table with crude field data as Supporting Information. See answers #3 and #13.

8. Please, check the final part of the conclusion (Either larger areas, lower pressures, or longer resting periods are necessary to exploit F. glabrescens sustainably) as it seems that this conclusion is not directly extracted from your data.

→ Thank you for noting that. We made a minor change in this statement (added words in bold), which we judge are enough to clarify our idea: “Our work demonstrates that the exploitation of F. glabrescens in southern Brazil, either at six months or one year intervals, leads to increased mortality rate, reduced mean size of ramnets, yields smaller than expected in previously unharvested areas, and a risk of even smaller yields if the exploitation goes on continuously. Either lower pressures over large areas or longer resting periods may be necessary to exploit F. glabrescens sustainably.” (See lines 270-274 in the annotated version). We understand that this conclusion follows from the results we got. The traditional practice of harvesting lianas at six months or one year intervals were not enough for the standing crop to recover, as demonstrated. Since the species reproduces from underground sprouts and ramets are exploited, lower pressure (less ramets cut per plant) over large areas could lead to the same desired crop while allowing for the plants to recover. We discussed the self-thining hypothesis as a potential explanation. Furthermore, longer resting periods are certainly another efficient alternative, even though we still don’t know how long this resting time should be at different pressures.

9. I also think that your study is rectricted to one location, and therefore you should mention any caveats to extrapolate your results to the management of the species in general.

→ We recognize this caveat and consigned a whole paragraph to deal with it. We made minor changes in the last statements of that paragraph (here in bold), recognizing that our results are confident at least to “the Atlantic, secondary forests of southern Brazil where most of the exploitation occurs. We added that we are confident that “Our results are also enough to rise concern for the need of careful monitoring while exploring non-timber forest products”. (see lines 267-269 in the annotated version).We hope this changes will be enough as a recognition, at the same time, of caveats and reaches of our job.

10. Please check in line 144 after packages if the name of one package is missing before the first reference.

→ Corrected. We added the name of the package – bbmle.

Reviewers' comments:

Reviewer's Responses to Questions

Comments to the Author

11. Is the manuscript technically sound, and do the data support the conclusions?

Reviewer #1: Yes

Reviewer #2: Yes

→ Thank you.

12. Has the statistical analysis been performed appropriately and rigorously? 

Reviewer #1: Yes

Reviewer #2: Yes

→ Thank you.

13. Have the authors made all data underlying the findings in their manuscript fully available?

Reviewer #1: Yes

Reviewer #2: No

→ We included a table with crude field data as Supporting Information. See answers #3 and #7

14. Is the manuscript presented in an intelligible fashion and written in standard English?

Reviewer #1: Yes

Reviewer #2: Yes

→ Thank you.

Review Comments to the Author

15. Reviewer #1: The Manuscript presents important elements to guide sustainable extractivism.

References are out of date

→ We did our best to review the most recent advancements in subjects relates with our rationale about the ecological effects of the exploitation of NTFP in general, and of lianas in particular. This is a poorly investigated subject. We did find a few interesting examples and an interesting new focus for the self-thinning principle. Based on the new references we updated the second and third paragraphs of the discussion section. See also the answer to questions #5, above, and #19, bellow.

16. Uneven number of parcels could not have affected the results? what was the authors' strategy to resolve this situation?

→ GLMM is robust to departures from common assumptions of Anova, as it uses a weighted mean instead of a grand mean. Having unbalanced groups has little influence on multilevel ML provided it is not really strong (Maas & Hox 2005 doi: 10.1027/1614-1881.1.3.86; Zuur et al 2010 doi: 10.1111/j.2041-210X.2009.00001.x). The major effect of unbalanced design is heterogeneity of variances. We checked heterogeneity as recommended by Grueber et al 2011 doi: 10.1111/j.1420-9101.2010.02210.x. VIF values were low. Power problems could became important in the case of strong unequal variances which was not the case. We thus understand that this is not a major issue.

Reviewer #2:

The manuscript “Survivorship and yield of a harvested population of Forsteronia glabrescens” submitted to Plos One describes the effect of harvesting on survivorship of a liana species in forest area of Brazil. It is well written, the ideas are clear, and the interest part is that encompass several subjects (ethnobotany, ecology, and forest management). The following observations may improve this manuscript:

17. First paragraph in the Introduction is too long. I suggest the authors to divide the ideas in two shorter paragraphs.

→ Done.

18. The second paragraph introduces the idea that the population of some liana species are declining due to overexploitation. It would be necessary to add a sentence about increasing liana abundance in the last decades:

Schnitzer, S. A., & Bongers, F. (2011). Increasing liana abundance and biomass in tropical forests: emerging patterns and putative mechanisms. Ecology letters, 14(4), 397-406.

Ingwell, L. L., Joseph Wright, S., Becklund, K. K., Hubbell, S. P., & Schnitzer, S. A. (2010). The impact of lianas on 10 years of tree growth and mortality on Barro Colorado Island, Panama. Journal of Ecology, 98(4), 879-887.

→ We addressed this issue in the statement “The high growth rates of some species, and the rapid colonization of forest edges and early successional phases cause them to be abundant elements in secondary forests and altered fragments [16, 17]. Many lianas have great capacity of regenerating after damage, so that a new branch quickly replaces a damaged branch [18]”. Schnitzer & Bongers, (2011) was already cited. We added to new references on the subject – da Cunha Vargas et al (2021) and Estrada-Villegas et al 2020. We added ‘increase’ in the statement that “Despite attributes favouring their increase in abundance in primary and secondary forests [16-18]. We didn’t want to expand to much the comments because the increase in abundance is not the focus of our research. See also question #5.

19. I recommend adding general information about liana demography in the Introduction. For example, relationships and trade-offs between growth and mortality, and shade-tolerance. Some papers on this regard:

→ We added more information about the self-thining effect, related with this subject. This is an interesting subject, since exploitation alleviates density, but the self-thining effect has not yet been addressed directly in exploitation studies. Gianoli et al (2010) was already cited. We added Ceballos and Malizia (2017). See also question #5.

20. It would be necessary to explain why these variables were selected: tree density, mean and maximum tree diameter at breast height (DBH), and tree DBH standard deviation, and how these variables are related to this particular liana species. There are some information on the literature? For example, tree DBH standard deviation represents the availability of trees of different sizes to climb to the canopy?

→ We expanded the statement declaring the reason to measure these variables in the methods section: “We used four variables describing the vegetation within each plot as random effects in order to control for the effect of potential differences in the forest stands on the abundance and fate of lianas and the availability of support trees and canopy cover: tree density, mean and maximum tree diameter at breast height (DBH), and tree DBH standard deviation. These DBH measures can account for potential effects of the variability in the availability of support trees and in the canopy cover, known to affect the abundance and growth of lianas [28, 29]”. We do not have predictions or hypotheses related with the random effects.

21. What was the minimum diameter for a liana to be considered? This information is needed in Methods.

→ We added in the methods that “We excluded from counting and harvesting young sprouts with length <1.0 m or diameter <0.2 mm, because they are not suitable for making handicrafts [26]”.

22. In Results change DAP by DBH.

→ done

23. In line 165, remove “The”.

→ done

24. Improve the quality of Figure 4.

→ done

25. Vines and ethnobotany are keywords but never mentioned in the text.

→ we selected synonyms as keywords in order to improve the chances of the paper reaching the interested audience. We preferred to standardize the words in the text.

26. It would be useful to add a Discussion on methodological issues involve in this study. Particularly, about the differences between doing this study in an area not exploited before (a relatively protected area in late succession) and not in areas that were harvested several times in the past. This probably affected the Results.

→ We declared that our objective was to reproduce “experimentally, under field conditions, the stem harvesting practised by the Kaingang natives to test if survivorship and yield differ between exploited and unexploited plots”. We rephrased the first sentence of the discussion in order to be more explicit about the reach of our results: “In this work we demonstrate that, compared with unexploited plots, the survivorship of Forsteronia glabrescens decreases in harvested plots, …”. The whole first paragraph focus on the comparison between exploited and unexploited stands.

---

## [Decision Letter · Decision Letter 1]

25 Mar 2022

PONE-D-21-21671R1Survivorship and yield of a harvested population of Forsteronia glabrescens .PLOS ONE

Dear Dr. Gaudagnin,

Thank you for submitting your manuscript to PLOS ONE. After careful consideration, we feel that it has merit but does not fully meet PLOS ONE’s publication criteria as it currently stands. Therefore, we invite you to submit a revised version of the manuscript that addresses the points raised during the review process.

I've received comments from three reviewers and I agree with their general assessments of your revised manuscript. Most importantly, you did a good job addressing the points raised by the previous reviewers. There are only a handful of things that you need to clarify further -- please see the reviewers' specific comments for guidance. Once you address these mostly minor comments and resubmit, I should be able to make a final editorial decision quickly.

We look forward to receiving your revised manuscript.

Kind regards,

Frank H. Koch, PhD

Academic Editor

PLOS ONE

Journal Requirements:

Reviewers' comments:

Reviewer's Responses to Questions

**Comments to the Author**

1. If the authors have adequately addressed your comments raised in a previous round of review and you feel that this manuscript is now acceptable for publication, you may indicate that here to bypass the “Comments to the Author” section, enter your conflict of interest statement in the “Confidential to Editor” section, and submit your "Accept" recommendation.

Reviewer #3: (No Response)

Reviewer #4: (No Response)

Reviewer #5: (No Response)

2. Is the manuscript technically sound, and do the data support the conclusions?

Reviewer #3: Yes

Reviewer #4: Yes

Reviewer #5: Yes

3. Has the statistical analysis been performed appropriately and rigorously? 

Reviewer #3: Yes

Reviewer #4: Yes

Reviewer #5: Yes

4. Have the authors made all data underlying the findings in their manuscript fully available?

Reviewer #3: Yes

Reviewer #4: Yes

Reviewer #5: No

5. Is the manuscript presented in an intelligible fashion and written in standard English?

Reviewer #3: Yes

Reviewer #4: Yes

Reviewer #5: Yes

6. Review Comments to the Author

Reviewer #3: This ms presents the results of an interesting, relevant and well conducted study on the sustainability of liana harvesting in Brazil. The experimental is appropriate, statistical analyses are sophisticated and correct, and results are interesting. I have some minor comments.

1. FIgures. In Figures 2 and 3, the unit of y-axis is incomplete. Is this annual survival and growth? I was confused by Figure 2: if for the first and second half-year period survival is expressed per half year, the numbers do not match, because a 65% survival in the semi-annual treatment in the first period could never lead to an ~80% survival for the full year. So my guess is that survival is annualized, but it's important to be clear about this, also in the text.

2. Negative radial growth. I was surprised to see negative radial growth rates. Are these measurement errors, or are they resulting from the fact that growth rate is based on the average diameter? In the latter case, I suggest to avoid confusion by calling this 'Change in mean stem diameter', instead of radial growth. IN any case, this needs to be better explained in the methods section.

Textual comments:

Line 251, remove "does"?

Line 261-262. Strange formulation; I suggest to change to: "Our results raise concerns about the sustainability of stem harvesting of our study species and we therefore call for continued monitoring of harvested populations from this species."

Reviewer #4: General considerations

Dear authors and editor in chief, the manuscript (PONE-D-21-21671) investigated how the exploitation influences survivorship, growth and yield of Forsteronia glabrescens, a liana from subtropical South America. The article is well written and address an important subject for the conservation of forests: how management practices affect species. Furthermore, the conclusions are supported by the gathered data and the influences of the weaknesses (one location and one-year long study) are properly discussed.

Authors did an excellent job addressing the suggestions of the previous referees. Thus, I made my few comments and suggestions based on the second version of the manuscript, as you will find below.

Introduction

• It would be nice to see some lines quantifying the importance of harvesting the Forsteronia glabrescens for the Kaingang people. For instance, how many families depends upon this harvesting? Alternatively, how much money they can make by selling the handicrafts? These informations can give a better picture about the importance of this species to the people who manage it.

Methods

• Please provide, if possible, any historic information about the forest used to install and evaluate the plots. For instance, is there any information about previous harvesting of lianas in the studied area?

• It is not clear to me how the authors compared the means of survivorship, radial growth and harvested length between treatments. For instance, survivorship in the control group is considered higher than the other treatments for the first period (lines 161-162), but based on what statistical test?

Results

• Figures 2 to 4 are not standing alone. What is the statistic depicted in the error bars? Confidence interval, standard deviation…? It seems to me that comparisons between means were made based on the error bars, but it remains unclear what statistic is depicted there and how it was estimated. Please provide this information in methods section.

• The scales of Fig 2 and 4 are wrong, because it varies from zero to one. If it is a percentage, it should vary from zero to 100.

• Please use points as decimal separator.

Reviewer #5: I believed the authors addressed properly the questions of the Reviewer 1. I read the manuscript again and found a few things that still need to be addressed. Most of them are related to unclear or confusing sentences and statements and are of a simple solution.

The paper focus on a very interesting subject, the impact of vines extraction on the vine population structure, survivorship, and yield. The authors made an experiment simulating two extraction intensities (similar to those practiced by traditional human exploitation of the vines) and evaluated the results for the vine population. Their results showed that both treatments (extraction once a year and extraction twice a year) do not allow enough time to population recover, resulting in a lower yield and reduction of the population.

The results and conclusions seem robust. Although, in my opinion, the number and size of the plots used as replicas could both be larger, the previous reviewers did not see any problem there and I agree that the results are solid. A larger sample possibly only leads to other significant results not found here, but certainly would not result in denying the significant results presented in the paper.

Minor comments:

Ln 51-52, 55-56, 224-225: When you say “compensatory and additive mortality”, I guess you mean that the recruitment after exploitation can have a compensatory or additive effect on the population. If so, it is not mortality that is compensatory or additive, but the recruitment as a response to the higher mortality generated by the exploitation.

Ln 58-60: Obscure sentence. What do you mean? Reduction of the population?

Ln 100-102: The different number of replicas by treatments could be a problem. I saw that the first reviewer also pointed to this problem. However, you are right; the statistical methods adopted are robust to overcome this shortcoming. However, it is hard to understand why you did that, so you need to justify this decision in the text.

Ln 134-136: Why is this in italic?

Ln 158-159: You need to tell the readers what it is DEC/14 and DEC/15.

Ln 162-163: Rewrite “About all the mortality after the initial harvest occurred in this period.”

Ln 163-164: These results contradict the graphic in Fig. 2. There, the annual harvest had less than 60% of survivorship. The semi-annual has survivorship higher than 72% after 1 yr.

Ln 164-165: Looking at the table, I cannot see this result “The cutting of stems reduced the survivorship in all treatments and periods.” The only thing I can see there is that the stem cutting affected the survivorship in the first period and, by consequence, in the whole period. No distinction is made for treatments.

Ln 166-167: But the maximum tree diameter had a negative relationship with survivorship. Do not omit results.

Ln 168: Table 1: This is hard to read. Some numbers seem to be out of place. Fix the table.

Ln 170-171: Fig. 2. You say that the y axis is %, but it is not. You have to multiply the number by 100 to get the %.

Ln 173: Checking the figure, the control group reduced in diameter, mainly during the first period.

Ln 176: “The radial growth increased with the cutting of stems” happened only in the first period.

Ln 189 Does “while the treatment groups measured 199.9 ± 130.8 cm” refer to both treatments together. Be clear about that.

Ln 190-191: “The growth in length of the harvested groups after one year was in average 40% lower than their initial length.” Where is this result shown?

Ln 191-192: I could not find this result in the table.

Ln 209: Exchange “Although” for “However”.

Ln 210-211: “Other cases of exploitation of perennial, understory NTFP showed comparable results to ours”. Too vague, how comparable? Maybe, "similar", but you should give the readers some parameters to conclude that.

Ln 211-216: You are repeating results presented previously. Right now you need to offer the reader some insights.

Ln 220-222: So, are you saying that the studied plants are investing more in below-ground parts? Can you prove it or you are only speculating? Your results show that the studied plant is not recovering enough in a single year, but saying that the plants are relocating energy to underground structures is a little too speculative, although an interesting point that could be suggested for further investigation.

Ln 234: The sentence “The density of F. glabrescens is similar to that found in other studies” is lost here.

Ln 242-244: Expand this idea. It seems an interesting one.

7. PLOS authors have the option to publish the peer review history of their article (what does this mean?). If published, this will include your full peer review and any attached files.

Reviewer #3: No

Reviewer #4: No

Reviewer #5: No

---

## [Author Response · Author response to Decision Letter 1]

1 Apr 2022

Below, we address each specific comment made by the referees, addressed in the revised manuscript “Survivorship and yield of a harvested population of Forsteronia glabrescens” 

Responses to comments:

Journal Requirements:

#1. Please review your reference list to ensure that it is complete and correct. If you have cited papers that have been retracted, please include the rationale for doing so in the manuscript text, or remove these references and replace them with relevant current references. Any changes to the reference list should be mentioned in the rebuttal letter that accompanies your revised manuscript. If you need to cite a retracted article, indicate the article’s retracted status in the References list and also include a citation and full reference for the retraction notice.

→ We added five references to account for the issues raised by the referees. Two to detail information about the Kaingang people and their usage of NTFPs in South Brazil; one as proof of the old-growth condition of the studied forest stand; and two to justify the robustness of the design and results despite unequal sample sizes. The included references are:

Freitas, AEC. Mrur Jykre – A cultura do cipó: territorialidade Kaigang na margem leste do Lago Guaíba, Porto Alegre, RS. Doctoral dissertation. Universidade Federal do Rio Grande do Sul. 2005. Available from: https://www.lume.ufrgs.br/handle/10183/14922

Fortes, PHR. Indígenas na cidade: Uma análise histórica e etnográfica da presença Kaigang em Curitiba. Doctoral dissertation. Universidade Federal do Paraná. 2020. Available from: https://acervodigital.ufpr.br/handle/1884/69422

Ribeiro, CV. Preceituação ecológica para a preservação dos recursos naturais na região da Grande Porto Alegre. Porto Alegre: Fundação Zoobotânica do Rio Grande do Sul; 1976.

Maas CJM, Hox JJ. Sufficient Sample Sizes for Multilevel Modeling. Methodology. 2005;1:86–92. https://doi.org/10.1027/1614-2241.1.3.86

Zuur AF, Ieno EN, Elphick CS. A protocol for data exploration to avoid common statistical problems: Data exploration. Methods in Ecology and Evolution. 2010;1:3–14. https://doi.org/10.1111/j.2041-210X.2009.00001.x

Reviewers' comments:

Review Comments to the Author

Reviewer #3:

#2. This ms presents the results of an interesting, relevant and well conducted study on the sustainability of liana harvesting in Brazil. The experimental is appropriate, statistical analyses are sophisticated and correct, and results are interesting. I have some minor comments.

→ Thank you for the recognition of our effort and outcome.

#3. Figures. In Figures 2 and 3, the unit of y-axis is incomplete. Is this annual survival and growth?

→ Fig.2 shows three sets of bars representing different periods – the first two of six months and the last one of one year. Because of that, we specified the length of each set in the subtitles instead of in the y-axis. We rephrased the subtitle to make it clearer. New subtitle reads: “Fig 2. Survivorship of Forsteronia glabrescens in two harvesting periods of six-months and after one year, comparing stems unharvested and submitted to harvesting once and twice a year.”

#4. I was confused by Figure 2: if for the first and second half-year period survival is expressed per half year, the numbers do not match, because a 65% survival in the semi-annual treatment in the first period could never lead to an ~80% survival for the full year. So my guess is that survival is annualized, but it's important to be clear about this, also in the text.

→ Thank you very much for finding that error in Fig. 2. We had pulled the wrong numbers when building Fig. 2 and apologize for that. The true figures of survival are 66% in the semi-annual treatment in the first period, 73% the second period, and 53% for the whole year. We replaced Fig. 2 and also corrected the values mentioned in the text. All interpretations remain the same, since they were based on the correct figures.

#5. Negative radial growth. I was surprised to see negative radial growth rates. Are these measurement errors, or are they resulting from the fact that growth rate is based on the average diameter? In the latter case, I suggest to avoid confusion by calling this 'Change in mean stem diameter', instead of radial growth. IN any case, this needs to be better explained in the methods section.

→ Thank you for the suggestion. We accepted it and changed the mentions to radial growth in the table 1 and along the text. We agree it will help to avoid confusion. We also explained in Methods that we calculated and then analyzed mean values. In order to standardize the wording and be more precise, we are using “Change in stem length” as well, instead of “stem growth”. See also issue #28.

#6. Textual comments:

Line 251, remove "does"?

→ Done.

#7. Line 261-262. Strange formulation; I suggest to change to: "Our results raise concerns about the sustainability of stem harvesting of our study species and we therefore call for continued monitoring of harvested populations from this species."

→ Done.

Reviewer #4: General considerations

#8. Dear authors and editor in chief, the manuscript (PONE-D-21-21671) investigated how the exploitation influences survivorship, growth and yield of Forsteronia glabrescens, a liana from subtropical South America. The article is well written and address an important subject for the conservation of forests: how management practices affect species. Furthermore, the conclusions are supported by the gathered data and the influences of the weaknesses (one location and one-year long study) are properly discussed.

Authors did an excellent job addressing the suggestions of the previous referees. Thus, I made my few comments and suggestions based on the second version of the manuscript, as you will find below.

→ Thank you for the recognition of our effort and outcome

Introduction

#9. It would be nice to see some lines quantifying the importance of harvesting the Forsteronia glabrescens for the Kaingang people. For instance, how many families depends upon this harvesting? Alternatively, how much money they can make by selling the handicrafts? These informations can give a better picture about the importance of this species to the people who manage it.

→ We included a statement regarding the number of Kaingangs in South Brazil and the importance of crafts and NTFPs. Unfortunately, there are no good figures. Our previous work [ref. 26] is the only one providing economic figures of the exploitation of lianas by Kaingangs. We cited two unpublished thesis in order to provide some figures.

Methods

#10. Please provide, if possible, any historic information about the forest used to install and evaluate the plots. For instance, is there any information about previous harvesting of lianas in the studied area?

→ We expanded the last sentence of the first paragraph of the Methods section, providing the figure that the old forest stand studied is out of direct human interference at least in the last 50 years.

#11. It is not clear to me how the authors compared the means of survivorship, radial growth and harvested length between treatments. For instance, survivorship in the control group is considered higher than the other treatments for the first period (lines 161-162), but based on what statistical test?

→ As stated in the Methods section, we used GLMM to analyze the data. Our statement about the significance of differences about treatments is based on the inclusion of the allocation of individuals in the experiment as an explanatory, categorical variable in the model (see page 6, lines 127-129 in the checked new version).

Results

#12. Figures 2 to 4 are not standing alone. What is the statistic depicted in the error bars? Confidence interval, standard deviation…? It seems to me that comparisons between means were made based on the error bars, but it remains unclear what statistic is depicted there and how it was estimated. Please provide this information in methods section.

→ Done. We rephrased the captions of figures 2, 3 and 4, and included the statement that the values represent means ± standard errors.

#13. The scales of Fig 2 and 4 are wrong, because it varies from zero to one. If it is a percentage, it should vary from zero to 100.

→ We are providing a new figure correcting the scale, as well as showing the data in a slightly different way to make it more clear – we are showing the change in the mean stem length of each period as well as the accumulated values at the end of the study. See also issues #26, #30, and #31.

#14. Please use points as decimal separator.

→ Checked.

Reviewer #5:

#15. I believed the authors addressed properly the questions of the Reviewer 1. I read the manuscript again and found a few things that still need to be addressed. Most of them are related to unclear or confusing sentences and statements and are of a simple solution. The paper focus on a very interesting subject, the impact of vines extraction on the vine population structure, survivorship, and yield. The authors made an experiment simulating two extraction intensities (similar to those practiced by traditional human exploitation of the vines) and evaluated the results for the vine population. Their results showed that both treatments (extraction once a year and extraction twice a year) do not allow enough time to population recover, resulting in a lower yield and reduction of the population. The results and conclusions seem robust. Although, in my opinion, the number and size of the plots used as replicas could both be larger, the previous reviewers did not see any problem there and I agree that the results are solid. A larger sample possibly only leads to other significant results not found here, but certainly would not result in denying the significant results presented in the paper.

→ Thank you for the recognition of our effort and outcome.

Minor comments:

#16. Ln 51-52, 55-56, 224-225: When you say “compensatory and additive mortality”, I guess you mean that the recruitment after exploitation can have a compensatory or additive effect on the population. If so, it is not mortality that is compensatory or additive, but the recruitment as a response to the higher mortality generated by the exploitation.

→ We prefer to keep the vernacular wording of the concept – compensatory and additive mortality – as defined in the mentioned literature. Hoping to clarify the idea, we rephrased the statements in lines 51-52, 55-56 to add the term “recruitment”. This term is already used in lines 224-225, so we left them unchanged. 

#17. Ln 58-60: Obscure sentence. What do you mean? Reduction of the population?

→ We rephrased the statement, focusing on the trade-off between recruitment and reproduction.

#18. Ln 100-102: The different number of replicas by treatments could be a problem. I saw that the first reviewer also pointed to this problem. However, you are right; the statistical methods adopted are robust to overcome this shortcoming. However, it is hard to understand why you did that, so you need to justify this decision in the text.

→ We included a statement about the unbalanced design in the paragraph commenting the weaknesses, in the Discussion. One plot of the annual harvest treatment had some labels lost or removed, probably by fauna, and was excluded from analyses. 

#19. Ln 134-136: Why is this in italic?

→ This was an error probably at the pdf building. The original text is not in italic.

#20. Ln 158-159: You need to tell the readers what it is DEC/14 and DEC/15.

→ We were unable to find this issue. Probably a problem only in the version submitted to the referee.

#21. Ln 162-163: Rewrite “About all the mortality after the initial harvest occurred in this period.”

→ Done.

#22. Ln 163-164: These results contradict the graphic in Fig. 2. There, the annual harvest had less than 60% of survivorship. The semi-annual has survivorship higher than 72% after 1 yr.

→ Corrected. Thank you for highlighting this error. We apologize for that. Actually, the numbers in the text were wrong. The correct values, as shown in Fig. 2, are 80.70% for the control group, 55.81% in the annual harvest and 53.19% in the semi-annual harvest. 

#23. Ln 164-165: Looking at the table, I cannot see this result “The cutting of stems reduced the survivorship in all treatments and periods.” The only thing I can see there is that the stem cutting affected the survivorship in the first period and, by consequence, in the whole period. No distinction is made for treatments.

→ We agree with the referee’s interpretation. Survivorship was affected, either positively or negatively, in all periods, but by different factors. Stem cutting only affected survivorship in the first period. We modified the statement about this result. We checked this subject in the discussion, but no conclusion needed to be reformulated.

#24. Ln 166-167: But the maximum tree diameter had a negative relationship with survivorship. Do not omit results.

→ Included. Although we had initially opted to not highlight this result in the text of Results section, we have discussed the issue in the Discussion section.

#25. Ln 168: Table 1: This is hard to read. Some numbers seem to be out of place. Fix the table.

→ Fixed.

#26. Ln 170-171: Fig. 2. You say that the y axis is %, but it is not. You have to multiply the number by 100 to get the %.

→ Done. See also issues #12, #30, and #31.

#27. Ln 173: Checking the figure, the control group reduced in diameter, mainly during the first period.

→ We included a statement in the results commenting that the control group showed minor variations in stem diameter along the year (maximum of 5%). These variations are caused by a few deaths and births (sprouts) along the period. We regard them as not relevant and did not discuss the issue.

#28. Ln 176: “The radial growth increased with the cutting of stems” happened only in the first period.

→ We included this detail in the corresponding phrase. See also issue #5.

#29. Ln 189 Does “while the treatment groups measured 199.9 ± 130.8 cm” refer to both treatments together. Be clear about that.

→ Stated explicitly, as asked.

#30. Ln 190-191: “The growth in length of the harvested groups after one year was in average 40% lower than their initial length.” Where is this result shown?

→ This result can be depicted from Fig. 4. We rebuilt this figure, changing the values shown from percentage change to percentage to facilitate the interpretation. We opted to not include another table to keep the manuscript short. See also issues #13 and #26.

#31. Ln 191-192: I could not find this result in the table.

→ This result can also be depicted from Fig. 4. See #30.

#32. Ln 209: Exchange “Although” for “However”.

→ Done. Thank you.

#33. Ln 210-211: “Other cases of exploitation of perennial, understory NTFP showed comparable results to ours”. Too vague, how comparable? Maybe, "similar", but you should give the readers some parameters to conclude that.

→ We expanded the statement.

#34. Ln 211-216: You are repeating results presented previously. Right now you need to offer the reader some insights.

→ Here we are highlighting the proofs for the conclusion that the population of F. grabrescens under-compensates the exploitation of stems at the experimented conditions. We included another phrase stating this. We believe that is essential to be explicit about all the effects of the harvesting on survivorship and stem diameter and length.

#35. Ln 220-222: So, are you saying that the studied plants are investing more in below-ground parts? Can you prove it or you are only speculating? Your results show that the studied plant is not recovering enough in a single year, but saying that the plants are relocating energy to underground structures is a little too speculative, although an interesting point that could be suggested for further investigation.

→ We believe we are being clear that the investment in bellow-ground organs is a discovery of others (reference [10]), under natural situations that, for the fate of the plant, are similar to a human exploitation (predation). Here we are pointing to an under-investigated subject in hope of stimulating other studies.

#36. Ln 234: The sentence “The density of F. glabrescens is similar to that found in other studies” is lost here.

→ We deleted the statement. It is just a minor comment.

#37. Ln 242-244: Expand this idea. It seems an interesting one.

→ Expanded. We added a comment about the probable population outcome when the relationship is facultative and services, instead o food, are demanded by the parasite, as the case of lianas and trees.

---

## [Editor Report · Decision Letter 2]

8 Apr 2022

PONE-D-21-21671R2Survivorship and yield of a harvested population of Forsteronia glabrescens .PLOS ONE

Dear Dr. Gaudagnin,

Thank you for submitting your manuscript to PLOS ONE. After careful consideration, we feel that it has merit but does not fully meet PLOS ONE’s publication criteria as it currently stands. Therefore, we invite you to submit a revised version of the manuscript that addresses the points raised during the review process. Thank you for completing another round of revisions. I appreciate your attention to the reviewers' comments. I believe the revised text is suitable for publication but there is a problem with the figures: what you uploaded as the new Fig. 2 was actually meant to replace Fig. 3. Consequently, there is no figure depicting survivorship (the intended subject of Fig. 2). This is an easy fix -- simply upload the correct versions of Figs. 2 and 3. Furthermore, you might change the legend labels in Fig. 1 to "Dec 2014" and Dec 2015" for added clarity, but this is a relatively minor detail.

We look forward to receiving your revised manuscript.

Kind regards,

Frank H. Koch, PhD

Academic Editor

PLOS ONE
---

## [Author Response · Author response to Decision Letter 2]

22 Apr 2022

In this version we only updated the figures, as requested. The text is the same one revised before as v.2.

---

## [Editor Report · Decision Letter 3]

4 May 2022

Survivorship and yield of a harvested population of Forsteronia glabrescens .

PONE-D-21-21671R3

Dear Dr. Gaudagnin,

We’re pleased to inform you that your manuscript has been judged scientifically suitable for publication and will be formally accepted for publication once it meets all outstanding technical requirements.

Kind regards,

Frank H. Koch, PhD

Academic Editor

PLOS ONE

Additional Editor Comments (optional):

Thank you for correcting the figures. Your manuscript is now suitable for publication.
---

## [Editor Report · Acceptance letter]

18 May 2022

PONE-D-21-21671R3 

Survivorship and yield of a harvested population of *Forsteronia glabrescens*. 

Dear Dr. Gaudagnin:

I'm pleased to inform you that your manuscript has been deemed suitable for publication in PLOS ONE. Congratulations! Your manuscript is now with our production department. 

Kind regards, 

on behalf of

Dr. Frank H. Koch 

Academic Editor

PLOS ONE